# Magnon interactions in a moderately correlated Mott insulator

Qisi Wang [1,2] ✉, S. Mustafi [2], E. Fogh [3], N. Astrakhantsev[2], Z. He[4,5],
I. Biało [2,6], Ying Chan[1], L. Martinelli[2], M. Horio [2], O. Ivashko [2], N. E. Shaik[3],
K. von Arx [2], Y. Sassa[7], E. Paris [8], M. H. Fischer [2], Y. Tseng [8],
N. B. Christensen [9], A. Galdi[10,11], D. G. Schlom [11,12], K. M. Shen [12,13],
T. Schmitt [8], H. M. Rønnow [3] & J. Chang [2] ✉

Quantum fluctuations in low-dimensional systems and near quantum phase transitions have significant influences on material properties. Yet, it is difficult to experimentally gauge the strength and importance of quantum fluctuations. Here we provide a resonant inelastic x-ray scattering study of magnon excitations in Mott insulating cuprates. From the thin film of $SrCuO_2$, single- and bi-magnon dispersions are derived. Using an effective Heisenberg Hamiltonian generated from the Hubbard model, we show that the single-magnon dispersion is only described satisfactorily when including significant quantum corrections stemming from magnon-magnon interactions. Comparative results on $La_2CuO_4$ indicate that quantum fluctuations are much stronger in $SrCuO_2$ suggesting closer proximity to a magnetic quantum critical point. Monte Carlo calculations reveal that other magnetic orders may compete with the antiferromagnetic Néel order as the ground state. Our results indicate that $SrCuO_2$−due to strong quantum fluctuations−is a unique starting point for the exploration of novel magnetic ground states.

One of the most studied electronic models is that of a two-dimensional square lattice[1]. At half-filling, when Coulomb interaction $U$ overwhelms greatly the kinetic energy scale $t$ the system crystallizes into an antiferromagnetically (AF) ordered Mott insulating state[2]. Layered cuprate systems such as $SrCuO_2$[3,4], $Nd_2CuO_4$, and $La_2CuO_4$ are archetypal examples[5]. The physics of these systems is well captured by the Hubbard model, from which an effective Heisenberg model can be generated to describe the AF ground state. In this strong-coupling limit, the spin exchange interactions are realized through virtual hopping processes. Upon down-tuning the interaction strength, the AF Mott state remains a theoretical ground state. However, in this limit, small perturbations (for example doping) can trigger new magnetic or metallic ground states[2,5]. Pushing the Mott state into this soft regime is therefore of great interest. When potential and kinetic energy scales are comparable, quantum fluctuations enter the problem. Many experimental and theoretical studies have addressed this intermediate region of $U/t$. Upon doping, high-temperature superconductivity has been found[5] and spin-liquid states are predicted in certain models[6].

[1]Department of Physics, The Chinese University of Hong Kong, Shatin, Hong Kong, China. [2]Physik-Institut, Universität Zürich, Winterthurerstrasse 190, CH-8057 Zürich, Switzerland. [3]Institute of Physics, École Polytechnique Fedérale de Lausanne (EPFL), CH-1015 Lausanne, Switzerland. [4]Institute of High Energy Physics, Chinese Academy of Sciences (CAS), 100049 Beijing, China. [5]Spallation Neutron Source Science Center (SNSSC), Dongguan 523803, China. [6]AGH University of Krakow, Faculty of Physics and Applied Computer Science, 30-059 Krakow, Poland. [7]Department of Applied Physics, KTH Royal Institute of Technology, SE-106 91 Stockholm, Sweden. [8]Swiss Light Source, Paul Scherrer Institut, CH-5232 Villigen PSI, Switzerland. [9]Department of Physics, Technical University of Denmark, DK-2800 Kongens Lyngby, Denmark. [10]Dipartimento di Ingegneria Industriale, Universita' degli Studi di Salerno, 84084 Fisciano, SA, Italy. [11]Department of Materials Science and Engineering, Cornell University, Ithaca, NY 14850, USA. [12]Kavli Institute at Cornell for Nanoscale Science, Ithaca, NY 14853, USA. [13]Department of Physics, Laboratory of Atomic and Solid State Physics, Cornell University, Ithaca, NY 14853, USA. ✉e-mail: qwang@cuhk.edu.hk; johan.chang@physik.uzh.ch

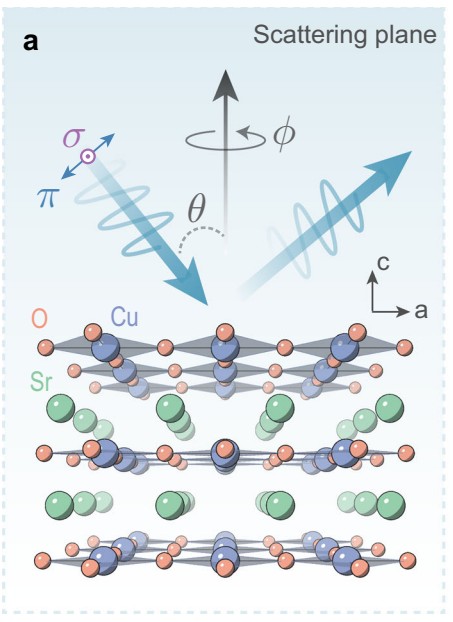

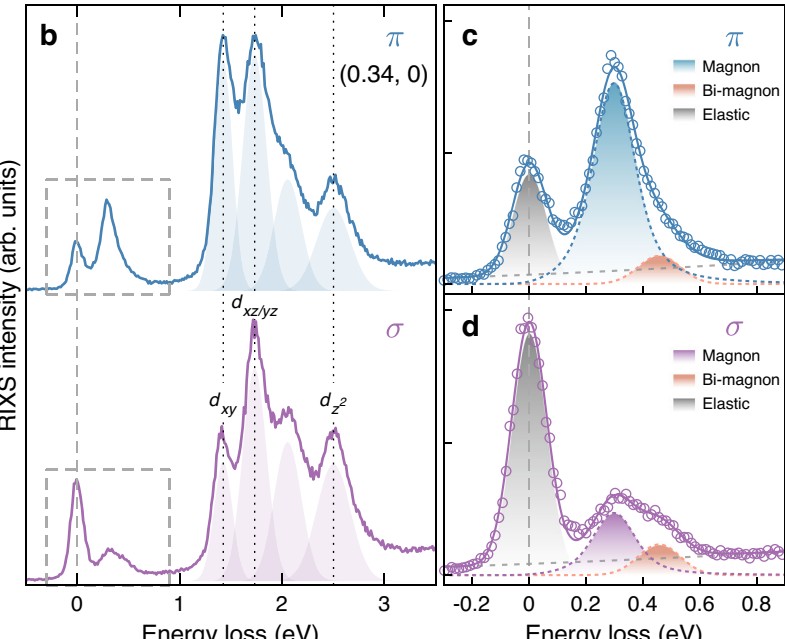

**Fig. 1 | RIXS on SrCuO₂ with different incident light polarizations. a** Crystal structure of SrCuO₂ and schematic illustration of the RIXS scattering geometry. Incident light, either linear horizontal ($\pi$) or vertical ($\sigma$), is directed to the film with variable angle $\theta$. **b** RIXS spectra at $\mathbf{Q} = (0.34, 0)$ measured with $\pi$ (blue line) and $\sigma$ (purple line) incident x-rays. Spectra are vertically shifted for clarity. Dotted lines mark the peak positions of the $dd$ excitations, determined from fitting with Gaussian components denoted by the shaded areas. **c, d** Zooms of the low-energy part (within the gray dashed boxes) of the spectra in (**b**). Solid lines are the sum of a four-component fit. Each component is indicated by dashed lines and shaded areas —see text and Supplementary Information for more details. Vertical dashed lines indicate the zero energy loss.

In spin-wave theory[7], quantum fluctuations describe the occupations of the bosonic modes as perturbations to the Néel state[8]. The ground state can be viewed as a Néel state with finite boson density. The corresponding elementary excitation should therefore be considered as a magnon renormalized by its higher-order expansions, i.e., the magnon-magnon interactions. On an experimental level, it has however been difficult to measure or gauge the strength of quantum fluctuations. As mentioned above, magnon excitations[9] of the Néel state should be influenced by quantum fluctuations. The magnon dispersion is described by $\hbar\omega = Z_c(k)\epsilon_k$, where $\epsilon_k$ is the "bare" magnon dispersion set by potential and kinetic energy scales, and $Z_c(k) = Z_c^0(1 + f_k)$ is the renormalization factor stemming from quantum fluctuations with $f_k$ being a momentum dependent function. In the strong coupling limit ($U/t \to \infty$) quantum fluctuations are suppressed implying $f_k \to 0$ and $Z_c(k) \approx 1.18$ is essentially momentum independent[10,11]. As quantum fluctuations grow stronger with gradually moderate values of $U/t$, the renormalization factor $Z_c(k)$ increases and acquires momentum dependence through a non-negligible $f_k$. This limit governed by quantum fluctuations is interesting as it may provide physics beyond the antiferromagnetic Néel state.

Conceptually, this moderate $U/t$ limit is complicated due to a multitude of comparable magnetic exchange interactions. Nearest and next-nearest neighbor exchange interactions are given by $J_1 = 4t^2/U$ and $J_2 = 4t^4/U^3$, in the projection of the Hubbard onto the Heisenberg model. In the moderate or weak interaction limit, higher-order exchange interaction terms are gaining prominence. The ring-exchange interaction $J_\square = 80t^4/U^3$ becomes a significant fraction of $J_1$ ($J_\square/J_1 = 20t^2/U^2$) and manifests by a magnon zone-boundary dispersion[12–17]. In this limit, higher-order hopping integral $t'$, can introduce new magnetic interaction term $J' = 4t'^2/U$ that further adds to enhance the zone boundary (ZB) dispersion. Within the Hubbard-Heisenberg model, the zone boundary dispersion, quantum fluctuations, and $Z_c$ correlate in the $U/t$ and $t'/t$ parameter space. In fact, the renormalization factor $Z_c$ gains its momentum dependence from the higher-order exchange interactions.

Enhanced quantum fluctuations may thus introduce new magnetic ground states and with that exotic magnonic quasiparticles[18,19]. It is thus interesting to study materials with significant higher-order exchange couplings. In the cuprates, $A$CuO₂ with $A$ = Sr, Ca has been studied with electron spectroscopy and resonant inelastic x-ray scattering (RIXS)[17,18,20] due to its large ring exchange interaction. Yet, no experiments have demonstrated the importance of magnon-magnon interactions and quantum fluctuations through direct measurements of a momentum dependent magnon renormalization factor.

Here we provide a RIXS study of SrCuO₂ (SCO) realized in thin-film format, which demonstrates a Mott-insulating nature by electron spectroscopy measurements[3,4]. Analysis of the RIXS spectra led us to derive the single- and bi-magnon dispersions. Starting from an effective Heisenberg representation of the Hubbard model, we show that the observed single-magnon dispersion is inconsistent with a constant $Z_c$ for reasonable values of kinetic energy scales. We thus conclude that, in SrCuO₂, quantum fluctuations are significantly influencing the magnon dispersion. Further, our analysis shows that the observed magnon dispersion is well described when introducing significant momentum dependence to $Z_c$ by evaluating the magnon-magnon interactions. This finding is further supported by comparing cuprate compounds with different correlation strengths. Our results thus provide a gauge for quantum fluctuations which are getting increasingly important as $U/t$ is reduced. Possible exotic magnetic ground states, emerging from quantum fluctuations are here explored by classical Monte Carlo calculations.

## Results

The crystal field environment around the copper site in SrCuO₂ is shown schematically in Fig. 1a. In contrast to, for example, La₂CuO₄ (LCO), no apical oxygen is present in SrCuO₂. Examples of Cu $L$-edge RIXS spectra, covering magnetic and $dd$ excitations, are shown in Fig. 1b. The absence of the apical oxygen in SrCuO₂ pushes the $d_{z^2}$ excitation well below the $t_{2g}$ excitations—as previously established in CaCuO₂[17,21]. The two excitations at 1.43 and 1.73 eV are assigned to the $d_{xy}$ and degenerated $d_{xz}$/

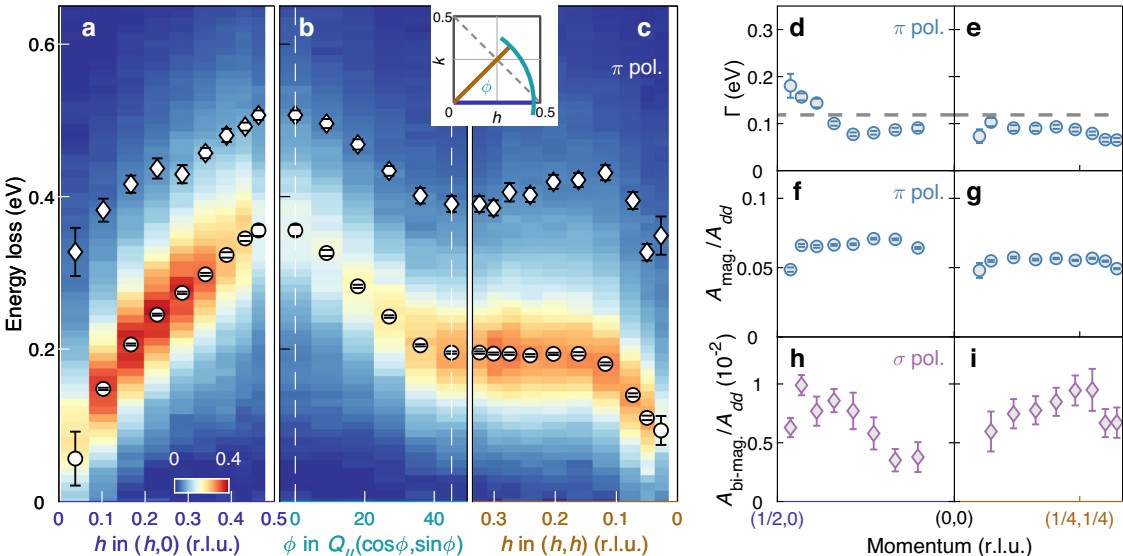

**Fig. 2 | Magnon and bi-magnon dispersions and spectral weights observed in SrCuO₂. a–c** RIXS intensity as a function of momentum and energy loss measured with $\pi$ polarized incident light, along three different directions in the reciprocal space as indicated by the solid color lines in the inset. Elastic and background scattering has been subtracted. Open circles and diamond points indicate respectively the magnon and bi-magnon pole position (see text for detailed description of the analysis). In (**b**) the in-plane momentum amplitude is $Q_{//} = 0.463$. **d, e** Single magnon inverse lifetime $\Gamma \sim \hbar/\tau$ along the $(h, 0)$ and $(h, h)$ directions. Horizontal dashed line indicates the applied energy resolution. Normalized single-magnon (**f, g**) and bi-magnon (**h, i**) spectral weight along the $(h, 0)$ and $(h, h)$ high symmetry directions. Error bars are determined from the fitting uncertainty.

$d_{yz}$ states, respectively. The origin of the additional peak at ~2.06 eV, which has also been observed in CaCuO₂[17,21], has been attributed to the incoherent component of the $d_{xz}/d_{yz}$ orbital excitations due to the coupling to magnetic excitations[22].

Magnetic excitations have been recorded systematically along the $(h, 0)$, $(h, h)$, and zone boundary (azimuthal $\phi$ rotation with a constant in-plane momentum amplitude $Q_{//}$) directions with both linear vertical ($\sigma$) and linear horizontal ($\pi$) incident light polarizations. A single-magnon excitation manifests clearly in the $\pi$ channel (see Fig. 1). When switching to $\sigma$ polarization, the single-magnon is suppressed as expected[23], and an excitation at higher energy appears. We interpret this as a magnetic continuum that in SrCuO₂ (and CaCuO₂) has a structure—sometimes referred to as a bi-magnon excitation[24–26]. In what follows, we extract the single-magnon and bi-magnon dispersions along the high symmetry directions.

We analyze the low-energy part of the RIXS spectra by considering four components that include elastic scattering (gray shaded area—enhanced in the $\sigma$ channel[23]), single- and bi-magnon (orange shaded area) excitations, and a smoothly varying background (gray dashed line). Elastic scattering is mimicked by a Gaussian function centered at zero energy loss. The energy width is slightly larger than the instrumental resolution due to unresolved phonon modes[27,28]. The single- and bi-magnon excitations are described respectively by a damped harmonic oscillator convoluted with the instrumental resolution and a Gaussian function. Background is modeled by a second-order polynomial. In grazing-exit geometry, RIXS cross section from the magnon (bi-magnon) is generally enhanced when using the $\pi$ ($\sigma$) polarized incident lights—as shown in Fig. 1c, d and Supplementary Fig. 1. We fit globally across the two light polarizations to extract the two magnetic contributions. The resulting single- and bi-magnon dispersions are plotted on top of the inelastic RIXS spectral weight ($\pi$ polarization) in Fig. 2a–c. Consistent with previous reports on CaCuO₂[17,18], a large zone boundary dispersion $E_{ZB} = \hbar\omega(\frac{1}{2}, 0) - \hbar\omega(\frac{1}{4}, \frac{1}{4})$ of the single magnon excitation is observed with an essentially non-dispersive section along the $(h, h)$ direction. Away from the Brillouin zone center, the bi-magnon dispersion $\hbar\omega_{bm}$ roughly mimics the single magnon

dispersion $\hbar\omega_{sm}$. At the zone boundary position $(\frac{1}{4}, \frac{1}{4})$, $\omega_{bm}/\omega_{sm} \approx 2$. This ratio however varies significantly along the high symmetry directions.

The fitting of the single- and bi-magnon excitations also provides information about spectral weight and quasiparticle lifetime. For most of the Brillouin zone, the energy width of the single-magnon is resolution-limited. However, around $(0.5, 0)$ spectral weight suppression and shorter single-magnon lifetimes are observed consistently with what has previously been reported in La₂CuO₄ and CaCuO₂[14,17,18].

## Discussion

The single-magnon dispersion of SrCuO₂ features two peculiar characteristics. A steep zone boundary dispersion is followed by a non-dispersive section along the $(h, h)$ direction. Magnon excitations of layered copper-oxides have been discussed via a Heisenberg Hamiltonian derived from the Hubbard model[2]. In the simplest form, the nearest-neighbor exchange interaction $J_1 = 4t^2/U$ is described through the Coulomb interaction $U$ and nearest-neighbor hopping integral $t$. The magnon dispersion is, in this limit, isotropic—given by $\hbar\omega = 2J_1(1 - ((\cos(Q_x) + \cos(Q_y))/2)^2)^{1/2}$. Early neutron scattering experiments on La₂CuO₄[12], however, revealed a zone boundary dispersion indicating the importance of higher-order exchange interaction terms. To account for this zone boundary term, a ring exchange interaction $J_\square \sim t^4/U^3$ was included to satisfactorily describe the observed magnon dispersion[12,14]. Later more detailed studies[29,30] included higher-order hopping terms, i.e., next, and next-next nearest-neighbor hopping integrals $t'$ and $t''$. This extended model, yields a magnon dispersion $\hbar\omega = Z_c(U, t, t', t'')\epsilon_k(U, t, t', t'')$, where $Z_c$ is a momentum dependent quantum renormalization factor and $\epsilon_k(U, t, t', t'') = \sqrt{A_k^2 - B_k^2}$ is the bare magnon dispersion with $A_k$ and $B_k$ determined by $U$, $t$, $t'$, and $t''$, as described in refs. 29,30. In the $U/t \to \infty$ limit, the magnon-magnon renormalization factor $Z_c$ is momentum independent, and $\epsilon_k(U, t, t', t'')$ has an analytic expression[16,31]. This expression has been used to fit the magnon dispersion of La₂CuO₄, with realistic values of $U$, $t$, $t'$, and $t''$. In particular, ratios of $t'/t \sim -0.4$ and $t''/t \sim 0.2$ are found consistent with density functional theory

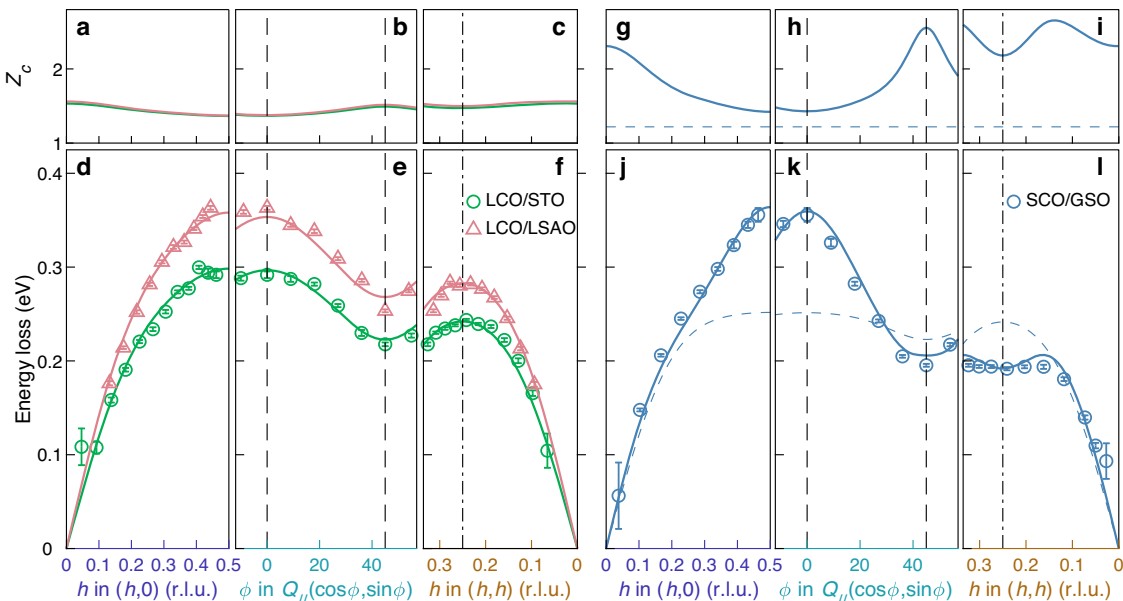

**Fig. 3 | Magnon dispersion and momentum dependence of quantum renormalization factor $Z_c$.** Bottom panels (**d–f**, **j–l**) display the magnon dispersion along the indicated momentum trajectories for LCO/STO (green), LCO/LSAO (pink), and SCO/GSO (blue). The solid lines are corresponding fits using a Hubbard model including higher-order terms (see text). The parameters extracted from the fits are listed in Table 1. Blue dashed lines mark the magnon dispersion obtained assuming a constant $Z_c = 1.219$[16,29], with $U = 2.15$ eV, $U/t = 6.25$, $t'/t = -0.4$, and $t''/t' = -0.5$. Error bars indicate one standard deviation. The in-plane momentum amplitude $Q_{//}$ takes 0.461, 0.444, and 0.463 for LCO/STO, LCO/LSAO, and SCO/GSO, respectively. Data on LCO/STO and LCO/LSAO are taken from ref. 16. Top panels (**a–c**, **g–i**) display the momentum dependence of the quantum fluctuation factor $Z_c$ obtained from fitting with the Hubbard model.

**Table 1 | Hubbard model parameters for magnons in La₂CuO₄, CaCuO₂ and SrCuO₂ films**

| Sample | $t$ [meV] | $U$ [eV] | $-t'/t$ | $-t''/t'$ | $U/t$ | $\bar{Z}_c$ | $\Delta Z_c/\bar{Z}_c$ [%] | $E_{ZB}/E(\frac{1}{2},0)$ [%] |
|---|---|---|---|---|---|---|---|---|
| LCO/STO | 408 | 3.38 | 0.4 | 0.5 | 8.33 | 1.45 | 11.6 | 18.9 |
| LCO/SLAO | 474.9 | 3.83 | 0.4 | 0.5 | 8.06 | 1.47 | 12.9 | 21.1 |
| CCO[17] | 498.7 | 3.39 | 0.42 | 0.5 | 6.80 | 1.89 | 57.3 | 40.1 |
| SCO/NGO | 425.6 | 2.66 | 0.4 | 0.5 | 6.25 | 1.91 | 64.5 | 47.3 |

Nearest, next-nearest neighbor hopping integral $t$, $t'$, and Coulomb interaction $U$ are obtained through self-consistently fitting the observed single magnon dispersion. Ratio between the next-next nearest hopping integral $t''$ and $t'$ was fixed to literature values[33,54,55]. Resulting electron correlation strength $U/t$ is anti-correlated with the zone boundary dispersion ratio $E_{ZB}/E(\frac{1}{2},0)$, average ($\bar{Z}_c$), and variation ($\Delta Z_c = \max(Z_c) - \min(Z_c)$) of $Z_c$ across the Brillouin zone.

calculations[32] and angle-resolved photoemission spectroscopy experiments[16,33].

For SrCuO₂, however, the constant $Z_c$ solution does not provide a satisfactory description of the observed single-magnon dispersion (see blue dashed lines in Fig. 3g–l). As La₂CuO₄ and SrCuO₂ share similar square lattice structures, similar values of $t'/t$ are expected. However, an unbiased fit yields unphysical values for the hopping parameters and physical sensible values provide poor fits. We are thus led to reject the initial ansatz $Z_c = Z_c^0(1 + f_k) \approx Z_c^0$ with $f_k \to 0$ and hence $Z_c$ being a momentum independent constant.

Consequently, we fit $\hbar\omega = Z_c(U,t,t',t'')\epsilon_k(U,t,t',t'')$ in a numerical self-consistent fashion. For La₂CuO₄, this methodology confirms that $Z_c$ is roughly constant (see Fig. 3a–c and Table 1) with marginal changes to $U,t,t',t''$ (compared to the constant $Z_c$ model[16]). However, for SrCuO₂, an entirely new solution emerges. Values of $t'/t$ and $t''/t$ comparable to La₂CuO₄ and a smaller $U/t$ now describes the magnon dispersion−see solid lines in Fig. 3 and Table 1. We stress that this new solution describes the observed dispersion using fewer fitting parameters as $Z_c$ is now given by $U,t,t',t''$. The moderate value of $U/t$ implies a magnon-magnon renormalization factor that is strongly momentum dependent (see Fig. 3g–i), i.e., $f_k$ in $Z_c = Z_c^0(1 + f_k)$ is no longer negligible. Our results thus indicate that quantum fluctuations have a significant impact on the magnon dispersion in SrCuO₂.

In Fig. 4, we plot the constant $\chi^2$ (goodness-of-fit) contour lines that encircle solutions that are within 10% of the minimum of $\chi^2$. With the currently available data, a rather broad set of parameters describe the magnon dispersion of La₂CuO₄. For SrCuO₂, however, $\chi^2$ has a unique well-defined minimum confined to a narrow region of the parameter space. In Table 1, the fitting values to describe the magnon dispersions for La₂CuO₄ and SrCuO₂ are listed. For SrCuO₂, these values represent a minimum in the $\chi^2$ function. For La₂CuO₄, we further constrain the solutions by fixing $t'/t = -0.4$. In Fig. 4, the modeled zone boundary dispersion is plotted as a function of $J_\square/J_1$ and $J'/J_1$. Within the same parameter space, the Brillouin zone average $\bar{Z}_c$ is shown. Generally, the model displays a correlation between magnon ZB dispersion and $\bar{Z}_c$. The Heisenberg-Hamiltonian projection from Hubbard model is breaking down in the limit where $\bar{Z}_c \gg Z_c^0$−that is when $(J_\square + J')/J_1$ is large.

Compared to La₂CuO₄, SrCuO₂ displays a larger ring exchange coupling $J_\square$. In fact, the fitting parameters obtained for SrCuO₂ are close to the limit where the Heisenberg representation of the Hubbard model breaks down. This limit is characterized by a complete suppression of the staggered magnetization and the imaginary solution of the magnon dispersion (Fig. 4). It has been reported that the absence of apical oxygen in cuprates leads to a decrease in the electronic correlation strength[34,35], which agrees with the observations here. To

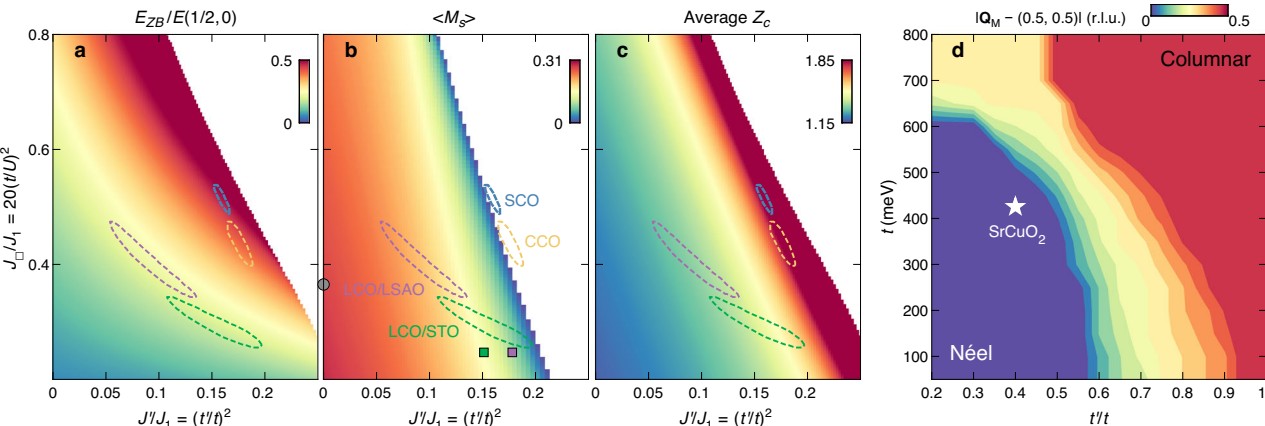

**Fig. 4 | Quantum renormalization effect within the Hubbard model. a, b** Zone boundary dispersion ratio $E_{ZB}/E(\frac{1}{2},0)$, renormalized staggered magnetization as a function of the exchange interactions $J_\square/J_1$ and $J'/J_1$. **c** Average renormalization factor $Z_c$ across the Brillouin zone for the same parameter space as in (**a, b**). Green, purple, yellow, and blue dashed circles indicate constant $\chi^2$ contour lines with solutions that are within 10% of the minimum of $\chi^2$ for LCO/STO, LCO/LSAO, CCO, and SCO, respectively. Gray, green, and purple filled symbols denote parameters

determined from the Hubbard model with constant $Z_c$ on respectively bulk LCO[12], LCO/STO, and LCO/LSAO[16]. Magnon dispersion of CCO is adapted from ref. [17]. Details of the analysis are described in the Supplementary Information. Empty areas in (**a, c**) and (**b**) indicate where the magnon dispersion becomes imaginary and the magnetization negative, respectively. **d** Magnetic ground state structure as a function of $t$ and $t'/t$ obtained from Monte Carlo calculations with $U = 2.66$ eV and $t''/t' = -0.5$ fixed. White pentagram marks the position of SCO.

corroborate our findings, we analyzed the magnon dispersion of CaCuO₂ (CCO), which is another infinite-layer cuprate compound with a large ring-exchange coupling[17]. As observed in ref. [17], the one-band Hubbard model with underestimated quantum renormalization fails to describe the magnon dispersion and yields an unrealistically small $U$ ($U/t = 4.9$). As shown in Fig. 4 and Supplementary Figs. 2 and 3, the magnon dispersions in both compounds are only well described when including substantial quantum corrections generated from magnon-magnon interactions. We thus demonstrate that SrCuO₂—a moderately correlated Mott insulator—hosts strong quantum fluctuations that can potentially stabilize ground states beyond the AF ordered Néel state. Enhancing further $J_\square/J_1$ or $J'/J_1$ would be of great interest to explore new quantum matter ground states.

To gain insight into the possible magnetic ground states when the Néel order breaks down by the increase of $t$ and $t'$, we perform Monte Carlo calculations using a Heisenberg Hamiltonian including the first-, second-, and third-nearest-neighbor, as well as a four-spin ring exchange coupling (see "Methods"). To compare with the experimental results on SrCuO₂, we fixed $U = 2.66$ eV and $t''/t' = -0.5$. Incommensurate magnetic orders characterized by a quartet of magnetic Bragg peaks around (0.5, 0.5), i.e., with wave vectors $\mathbf{Q}_M = (0.5 \pm \delta, 0.5)$ and $(0.5, 0.5 \pm \delta)$, or $\mathbf{Q}_M = (0.5 \pm \delta/\sqrt{2}, 0.5 \pm \delta/\sqrt{2})$ are found in the parameter space between the antiferromagnetic Néel and columnar orders (see Supplementary Fig. 4 for examples of the calculated spin structure factor). We plot in Fig. 4d the distance $\delta$ between $\mathbf{Q}_M$ and the Néel wave vector (0.5, 0.5) as a function of $t$ and $t'/t$. While SrCuO₂ is in the Néel AF ordered state, it is located not far from incommensurate magnetic ordered phases which can be reached by increasing $t$. Further increase of $t'/t$ enhances the next-nearest neighbor coupling and stabilizes the columnar antiferromagnetic order. This tuning can be potentially realized by strain application with different substrates[16,36]. Explorations on possible strain or pressure induced quantum critical behavior would provide more insights into the nature of the magnetic ground states. Note that recent theoretical and numerical works on the two-dimensional Hubbard model have shown that under small doping, the Néel order becomes unstable and replaced by other magnetic orders[37–41]. A spin-charge stripe state with an incommensurate ordering wave vector has also been experimentally established in underdoped cuprates[42–44]. We point out that the classical Monte Carlo simulations do not capture the

quantum nature of the problem, but they show the parameter space, where the Néel state is expected to break down. When antiferromagnetic order is suppressed by the strong quantum fluctuations—stemming from magnon-magnon interactions—near the phase boundary, superconductivity could be potentially enhanced. It would therefore be of great interest to study how the different magnetic ground states influence superconductivity upon doping. Future theoretical studies including extended dynamical mean-field theory calculations on the spin excitations[45–49]—beyond the scope of the current work—could offer more insights into the relationship between the quantum fluctuations and the degenerate ground states near the critical region. On the experimental front, such quantum effects could also be addressed by comparative RIXS measurements at a higher temperature ($T \sim 0.1J$), which call for future exploration.

## Methods
### Film growth
High-quality SrCuO₂ and La₂CuO₄ thin films were grown using molecular beam epitaxy (MBE). The SrCuO₂ (-15 nm) film is deposited on a (110) GdScO₃ (GSO) substrate[4,50], and La₂CuO₄ films are grown on (001) SrTiO₃ (STO) and LaSrAlO₄ (LSAO) substrates, with a thickness of 7–8 and 18–19 nm, respectively[16].

### RIXS experiments
Cu $L_3$-edge RIXS experiments were carried out at the ADRESS beamline[51,52], of the Swiss Light Source (SLS) synchrotron at the Paul Scherrer Institut. All data were collected at base temperature (-20 K) under ultrahigh vacuum (UHV) conditions, $10^{-9}$ mbar or better. RIXS spectra were acquired in grazing-exit geometry with both linear horizontal ($\pi$) and linear vertical ($\sigma$) incident light polarizations with a fixed scattering angle $2\theta = 130°$. The two-dimensional nature of the system ensures that the out-of-plane dependence of the magnon dispersion is negligible, as confirmed in a recent RIXS study on CaCuO₂[18]. The energy resolution, estimated by the full-width-at-half-maximum of the elastic scattering from an amorphous carbon sample, is 118.5 meV at Cu $L_3$ edge (-931.5 eV). Momentum transfer is expressed in reciprocal lattice units (r.l.u.) using a pseudo-tetragonal unit cell with $a = b = 3.97$ Å and $c = 3.4$ Å for SrCuO₂, $a = b = 3.8$ Å and $c = 13.2$ Å for La₂CuO₄. RIXS intensities are normalized to the weight of $dd$ excitations[53].

## Modeling of magnon

Fitting of the magnon dispersion was done by calculating self-consistently $Z_c$ and $\epsilon_{\mathbf{k}}$ in $\hbar\omega = Z_c(U,t,t',t'')\epsilon_{\mathbf{k}}(U,t,t',t'')$. We used $t''/t' = -0.5$[54]. Fitting parameters $U$ and $t$ were obtained by minimizing $\chi^2$. The staggered magnetization is calculated by $\langle M_s^\dagger \rangle = (S + \frac{1}{2} - \frac{2}{N}\sum_{\mathbf{k}} \frac{4_{\mathbf{k}}}{2\epsilon_{\mathbf{k}}})(1 - 8t^2/U^2)$, where $N$ is the number of spin on the square lattice. Our code is available upon request.

## Monte Carlo simulations

Monte Carlo simulations were carried out using the Heisenberg Hamiltonian projected from the Hubbard model, taking into consideration the leading contributions of $t'$ and $t''$:

$$\hat{\mathcal{H}} = J\sum_{\langle i,j\rangle} \mathbf{S}_i \cdot \mathbf{S}_j + (J_2 + J')\sum_{\langle i,i'\rangle} \mathbf{S}_i \cdot \mathbf{S}_{i'}$$
$$+ (J_3 + J'')\sum_{\langle i,i''\rangle} \mathbf{S}_i \cdot \mathbf{S}_{i''} + J_\square \sum_{\langle i,j,k,l\rangle} \Big[ \big(\mathbf{S}_i \cdot \mathbf{S}_j\big)\big(\mathbf{S}_k \cdot \mathbf{S}_l\big) \quad (1)$$
$$+ \big(\mathbf{S}_i \cdot \mathbf{S}_l\big)\big(\mathbf{S}_k \cdot \mathbf{S}_j\big) - \big(\mathbf{S}_i \cdot \mathbf{S}_k\big)\big(\mathbf{S}_j \cdot \mathbf{S}_l\big)\Big],$$

where $J = J_1 - \frac{24t^4}{U^3}$, $J_1 = \frac{4t^2}{U}$, $\frac{J_2}{J_1} = \frac{J_3}{J_1} = \left(\frac{t}{U}\right)^2$, $\frac{J'}{J_1} = \left(\frac{t'}{t}\right)^2$, $\frac{J''}{J_1} = \left(\frac{t''}{t}\right)^2$ and $\frac{J_\square}{J_1} = 20\left(\frac{t}{U}\right)^2$. The first three sums count respectively nearest, next-nearest, and next-next-nearest neighbor using indices $\langle i,j\rangle$, $\langle i,i'\rangle$ and $\langle i,i''\rangle$. The last sum counts around the squares following the clockwise direction. Simulations were performed for a temperature $T = 0.1$ K on a sheet of $50 \times 50$ unit cells in the $(a,b)$-plane, i.e., 2500 magnetic sites with classical spin $S = \frac{1}{2}$. For each set of input values, $(U,t,t',t'')$, the simulation ran for $10^7$ Monte Carlo steps with random starting configurations. The Monte Carlo calculation code is available upon request.

## Data availability

Data supporting the findings of this study are available from the corresponding authors upon request. Source data are provided with this paper.

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

## Acknowledgements

We thank André-Marie Tremblay, Michel Gingras, Takami Tohyama, and Krzysztof Wohlfeld for insightful discussions. RIXS measurements were performed at the ADRESS beamline of the SLS at the Paul Scherrer Institut, Villigen PSI, Switzerland. We thank the ADRESS beamline staff for technical support. M.H., O.I., K.v.A., N.A., E.P., Y.T., T.S. and J.C. acknowledge support by the Swiss National Science Foundation through grant numbers PP00P2_176877, 200021_188564, CRSII2_160765/1 and BSSGIO_155873. Q.W. is supported by the Research Grants Council of Hong Kong (ECS No. 24306223), and the CUHK Direct Grant (4053613 and 4053671). Work at Cornell was supported by the Air Force Office of Scientific Research grant number FA9550-21-1-0168 and the National Science Foundation through DMR-2104427 and DMR-2039380. I.B. and L.M. acknowledge support from the Swiss Government Excellence Scholarship under project numbers ESKAS-Nr: 2022.0001 and ESKAS-Nr: 2023.0052. E.F. and H.M.R. acknowledge support by the European Research Council through the Synergy network HERO (Grant No. 810451), and the Swiss National Science Foundation through Project Grant No. 188648. N.B.C acknowledges support from the Danish National Council for Research Infrastructure (NUFI) through the ESS-Lighthouse Q-MAT. Y.S. acknowledges funding from the Wallenberg Foundation (KAW 2021.0150).

## Author contributions

Q.W. and J.C. conceived the project. A.G. grew and characterized the $SrCuO_2$ film with advice from D.G.S. and K.M.S. Q.W., M.H., O.I., K.v.A., Y.S., E.P., Y.T. and T.S. carried out the RIXS experiments. Q.W., S.M., N.A. and Y.C. analyzed the RIXS data with the help from I.B., L.M., N.E.S., M.H.F. and N.B.C. E.F., Z.H. and H.M.R. performed the Monte Carlo calculations. All authors contributed to the writing of the manuscript.

## Competing interests

The authors declare no competing interests.
