## [Peer Review File · Nature Communications]

REVIEWER COMMENTS

Reviewer #1 (Remarks to the Author):

In the work "Quantum Fluctuations in a Weakly Correlated Mott Insulator," the authors investigate single- and bi-magnon excitations in a thin film cuprate SrCuO₂ by means of the resonant inelastic X-ray scattering (RIXS) technique." To extract magnon dispersions, the low-energy part of the RIXS spectra is fitted by considering four components: elastic scattering, single- and bi-magnon excitations, and a smoothly varying background. The obtained result is analyzed based on the Heisenberg model. Furthermore, quantum Monte Carlo calculations for this model are performed to investigate the ground state magnetic order.

The primary finding of this study is that the magnon dispersion in SrCuO₂ is substantially different from the one observed in the other cuprate compound, namely La₂CuO₄. In particular, the authors show that the magnon dispersion cannot be reproduced by rescaling the bare magnon dispersion with a constant renormalization factor; instead, it requires consideration of a momentum-dependent renormalization. This finding is attributed to the effect of strong quantum fluctuations.

The manuscript presents a careful experimental study and is written clearly. However, I have several questions concerning the interpretation of the results that currently prevent me from recommending this work for publication in the Nature Communications journal.

The authors entitle their work "Quantum Fluctuations in a Weakly Correlated Mott Insulator." I would suggest reconsidering this title because the Mott insulating state is formed as a result of strong local electronic correlations, making this phrasing sound confusing.

In the title and introduction, the authors emphasize that they are dealing with Mott insulating cuprates. However, no confirmation of this fact can be found in the manuscript. On the contrary, the estimated value of the local Coulomb interaction U , as specified in Table I, is only approximately 6 times smaller than the nearest-neighbor hopping amplitude t . This implies that the system is more of a correlated metal than a Mott insulator. It would be beneficial if the authors could provide the electronic spectral function to support their statement that the system lies in the Mott insulating phase.

In my opinion, the authors place too much emphasis on the novelty of a possible incommensurate magnetic order in SrCuO₂. In fact, it has already been demonstrated in several theoretical works [PRB, 101, 165142 (2020); PRB 108, 035139 (2023); PRX 13, 011007 (2023); arXiv:2209.09237] that the magnetic ground state in the 2D Hubbard model corresponds to stripe order. Furthermore,

incommensurate magnetic ordering has already been observed in cuprates upon hole doping (see, e.g., [New J. Phys. 11 115004 (2009)])”

The authors attribute the discrepancy between the magnon dispersion and the bare magnon dispersion multiplied by a constant renormalization factor to the effect of strong quantum fluctuations. In my opinion, this statement lacks sufficient justification in the manuscript. In this work, the bare magnon dispersion is derived from the Heisenberg model, with exchange interaction values taken in the so-called strong coupling limit ($t \ll U$). This approximation appears valid for the La₂CuO₄ compound, although the U/t ratio in this material is approximately equal to 8 (Table I), which already doesn't strictly correspond to the $U/t \gg 1$ regime. The local Coulomb interaction in SrCuO₂ is even smaller ($U/t = 6.25$), making the strong coupling approximation for exchange interactions definitively invalid in this case. Firstly, in a strongly correlated metal regime, the exchange interactions have a much more complex form (see, e.g., [PRB 94, 115117 (2016); PRL 121, 037204 (2018); PRB 105, 155151 (2022)]). Secondly, the perturbation expansion in terms of t/U does not apply here, so one would likely need to account for rather long-range and multi-spin exchange interactions (see, e.g., discussions in [RMP 95, 035004 (2023)]). In this regard, I am wondering if the discrepancy between the fit and the actual data simply arises from an inappropriately considered Heisenberg model, unrelated to quantum fluctuations. In my opinion, a clear demonstration that the observed effect is indeed due to quantum fluctuations would be the fact that the magnon dispersion is not reproduced by solving the classical Heisenberg model but is instead reproduced by solving the corresponding quantum Heisenberg model. Alternatively, the authors could obtain magnon dispersions by directly solving the Hubbard model, which, for a single-band case, can be done using diagrammatic extensions of the dynamical mean-field theory [RMP 90, 025003 (2018)]. The spectral function for magnetic fluctuations can be obtained by calculating the spin susceptibility (see, e.g., [PRL 107, 137007 (2011); EPL 122, 57001 (2018); npj Quant. Mater. 3, 54 (2018); Commun. Phys. 2, 163 (2019)]). The results can then be compared to the magnon dispersion deduced from the Heisenberg model with proper exchange interactions.

Small points:

It would be helpful if the authors mention the energies or colors corresponding to the four components when discussing the low-energy part of the RIXS spectra (first paragraph on the right side of page 2). Otherwise, it is difficult to identify the bi-magnon peak in Fig. 1.

The estimated value of the next-nearest-neighbor hopping $|t'/t|=0.4$ is rather large compared to what is usually used to describe cuprates ($|t'/t|_{\text{max}} = 0.3$). It would be helpful if the authors could compare the obtained model parameters to those obtained in density functional theory (DFT) calculations.

In the Heisenberg model, the authors consider only particular types of exchange interactions, neglecting, e.g., the three-spin interaction, which should be non-zero in the case of an incommensurate spin order. Is there a specific reason for this, and can this interaction be added to the model?"

Reviewer #2 (Remarks to the Author):

In their manuscript, Wang et al present a RIXS investigation of the infinite layer SrCuO₂ (SCO), a Mott insulator distinguished in the cuprate realm by the absence of apical oxygen. The authors specifically focus on magnetic excitations, measuring them at different crystal orientations (H0, HH, and azimuthal). It is observed that the measured energy dispersion in q of the magnetic excitations cannot be fitted using the Hubbard model. This model, which effectively sets potential and kinetic energy scales, aligns well with La₂CuO₄, a cuprate parent compound characterized by the presence of apical oxygen. According to the authors, this discrepancy can be reconciled by considering a momentum-dependent normalization factor Z , intricately linked to quantum fluctuations. The conclusions drawn from the fit parameters, notably the substantial value of the exchange coupling, lead the authors to assert that in this system, the antiferromagnetic ordered Néel state is in close proximity to breakdown conditions. This implies the potential stabilization of novel magnetic ground states under extreme conditions, a hypothesis explored further through Monte Carlo simulations.

The significance of the study is underscored by the intense investigation of infinite-layer cuprates in recent times, revealing unexpected phenomena that could contribute to unraveling the physics behind HTS cuprates.

The dataset underpinning the paper is commendable in terms of quality, and the authors extraction of the magnon dispersion is rather convincing. The different scenarios occurring for SrCuO₂ and La₂CuO₄ are evident. On the contrary, the connection between the unconventional magnon dispersion of SCO and the presence quantum fluctuations looks weaker. The argument relies exclusively on the chosen fit, wherein the relationship between "measured" and "expected" magnon energy is mediated by a quantum-fluctuation-driven normalization factor. While there is a temptation to draw the conclusion that quantum fluctuations are steering the unconventional behavior, the current evidence suggests more of a suggestive hint than an established fact.

To substantiate the assertion of the presence of quantum fluctuations in the system, a more thorough comparison with previously collected similar data and an increased dataset are necessary. The manuscript conveys a strong statement on this matter, commencing from the title, and a more robust foundation is needed.

A more detailed comparison with previously measured existing data and a more complete dataset would be required to claim the presence of quantum fluctuations in the system, a rather strong statement appearing in the manuscript starting from the title.

Going into specifics, the observed behavior presented by the authors in Fig. 2 bears a striking resemblance to that previously documented in CCO in Ref. 13. There, the magnon dispersion could not be fitted using the nearest-neighbour Heisenberg model, differently than LCO. However, the fit is rather accurate using a one-band Hubbard model or, even better, the phenomenological linear spin-wave Heisenberg model with four nearest-neighbour coupling parameters. Notably, already with a one-band Hubbard model, the discrepancy between LCO and CCO can be reconciled by using a q -independent normalization factor Z and a ring exchange value in CCO, about 5 times larger than in LCO. This prompts a series of questions: What distinguishes the fit in Ref. 13 from the fit employed by the authors of the present manuscript? How do the authors motivate their choice with respect to what has been done in prior RIXS experiments? Is there a compelling physical rationale justifying their selected approach as the most appropriate?

On the experimental front, additional data may be indispensable to adequately substantiate the claim of quantum fluctuations in the system. For instance, if these fluctuations bear a quantum nature, shouldn't we observe a temperature dependent trend? A robust examination of the temperature evolution of the measured magnon dispersion would serve as a stringent test. One would reasonably expect that, with increasing temperature, the "measured" and the "bare" magnon energy should converge, signifying the waning influence of quantum fluctuation contributions as the temperature is raised.

In general, the manuscript would benefit from an improved discussion about the physics entailing these quantum fluctuations. Examining, from instance, their expected evolution with temperature and doping, and exploring their possible connection to a quantum critical point, would significantly enrich the depth and breadth of this manuscript.

Beside this main concern, I list in the following other minor issues I see:

- 1) Upon evaluating the pole positions of magnon energy in Figure 2c derived from the fit shown in Fig. 1, the dispersion in SrCuO₂ seems notably flat at q values above 0.15 rlu along the (h, h) direction. This region constitutes the primary point of qualitative disagreement with the Hubbard fit employing a constant Z (dashed line in Fig. 3l), predicting an upward curvature in that q range. However, this plateau is not evident in the bimagnon dispersion within the same q range, despite the latter mimicking the magnon dispersion at other q values, as explicitly noted by the authors. In that q range, the bimagnon dispersion displays an upward curvature, consistent with the behavior observed in the magnon data presented in the map of Fig. 2c. It is noteworthy that the Hubbard fit with higher-order terms, utilized for assessing quantum fluctuations, exhibits a mild, albeit opposite behavior in that q range, manifesting a softening of magnonic energy versus q . A more detailed explanation from the authors elucidating the reasons behind these incongruences is needed.

2) The magnon dispersion is measured at a constant 2θ , i.e., at variable L . Do the authors expect the L dependence being non-influent at all, Considering the 2D nature of the CuO₂ planes housing magnetic excitations and the infinite layer structure of SCO? A discussion on this point would be useful.

3) Discussing the $(h,0)$ and (h,h) directions in terms of antinodal and nodal directions is always a bit misleading in this context, as these terms evoke aspects of the material electronic structure and the shape of the superconducting gap.

4) Figure 1c and 1d suggest that the second-order polynomial approximation used in the fit to eliminate the background, primarily originating from the electron-hole continuum, is somewhat abrupt. It appears nearly energy-independent, differing from zero both at zero energy loss and in the antistokes region. This could potentially impact the determination of the pole of the bimagnon excitations, whose height is approximately as that the background, or at least significantly increasing its uncertainty. A detailed explanation or refinement of this aspect by the authors is encouraged.

5) In Figure 2a and 2b, there is a conspicuous suppression of spectral weight at $(0.5,0)$. Given this observation, it is perplexing why the error bars determined for the pole positions of magnons and bimagnons appear almost unaffected by such a signal drop.

6) In the methods, for clarity the instrumental energy resolution should be specified in terms of FWHM.

7) An even minor color coding issue is noted in Figures 3 and 4. The lines, data points, and dashed circles referring to LCO/STO and LCO/LSAO are respectively represented in green and pink in both figures. Conversely, for SCO, they are blue in Figure 3 and orange in Figure 4. To maintain consistency, it is suggested that one color be chosen for both figures in this instance as well.

RESPONSE TO REVIEWER COMMENTS

Reviewer #1 (Remarks to the Author):

In the work “Quantum Fluctuations in a Weakly Correlated Mott Insulator,” the authors investigate single and bi-magnon excitations in a thin film cuprate SrCuO₂ by means of the resonant inelastic X-ray scattering (RIXS) technique.” To extract magnon dispersions, the low-energy part of the RIXS spectra is fitted by considering four components: elastic scattering, single- and bi-magnon excitations, and a smoothly varying background. The obtained result is analyzed based on the Heisenberg model. Furthermore, quantum Monte Carlo calculations for this model are performed to investigate the ground state magnetic order.

The primary finding of this study is that the magnon dispersion in SrCuO₂ is substantially different from the one observed in the other cuprate compound, namely La₂CuO₄. In particular, the authors show that the magnon dispersion cannot be reproduced by rescaling the bare magnon dispersion with a constant renormalization factor; instead, it requires consideration of a momentum-dependent renormalization. This finding is attributed to the effect of strong quantum fluctuations.

The manuscript presents a careful experimental study and is written clearly. However, I have several questions concerning the interpretation of the results that currently prevent me from recommending this work for publication in the Nature Communications journal.

Authors: We thank the referee for the precise summary of our work and appreciate the positive comments on our manuscript. Below, we provide a point-by-point reply to the comments and suggestions made by the referee.

The authors entitle their work “Quantum Fluctuations in a Weakly Correlated Mott Insulator.” I would suggest reconsidering this title because the Mott insulating state is formed as a result of strong local electronic correlations, making this phrasing sound confusing.

Authors: We agree that “weakly correlated” and “Mott insulator” without context can appear confusing. Therefore, we have changed the title to “Magnon Interactions in a Moderately Correlated Mott Insulator”.

In the title and introduction, the authors emphasize that they are dealing with Mott insulating cuprates. However, no confirmation of this fact can be found in the manuscript.

Authors: The referee makes a valid point. In the revised version of the manuscript, we are now directly mentioning compound examples. That is, we are mentioning parent cuprate compounds (including SrCuO₂) that are Mott insulating (demonstrated in the cited literature).

On the contrary, the estimated value of the local Coulomb interaction U , as specified in Table I, is only approximately 6 times smaller than the nearest-neighbor hopping amplitude t . This implies that the system

is more of a correlated metal than a Mott insulator. It would be beneficial if the authors could provide the electronic spectral function to support their statement that the system lies in the Mott insulating phase.

Authors: A zero-order rule of thumb is that the Mott insulating state emerges when $U/t > 8$. That is when the Coulomb interaction U is larger than the band width $8t$. Now, higher-order hoppings might change this and so the $U/t > 8$ rule of thumb should be taken with a grain of salt. On an experimental level, La_2CuO_4 is certainly a Mott insulator. Yet, parametrization of the magnon dispersion yields values of $U/t < 8$ [PRL **86**, 5377 (2001)].

In my opinion, the authors place too much emphasis on the novelty of a possible incommensurate magnetic order in SrCuO_2 . In fact, it has already been demonstrated in several theoretical works [PRB, 101, 165142 (2020); PRB 108, 035139 (2023); PRX 13, 011007 (2023); arXiv:2209.09237] that the magnetic ground state in the 2D Hubbard model corresponds to stripe order. Furthermore, incommensurate magnetic ordering has already been observed in cuprates upon hole doping (see, e.g., [New J. Phys. 11 115004 (2009)]).”

Authors: We thank the referee for pointing out these recent theoretical studies, which we now cite in the revised manuscript. In the revised abstract, we now only mention the potential competing magnetic orders for the ground state without stating that they are “exotic incommensurate”.

The authors attribute the discrepancy between the magnon dispersion and the bare magnon dispersion multiplied by a constant renormalization factor to the effect of strong quantum fluctuations. In my opinion, this statement lacks sufficient justification in the manuscript. In this work, the bare magnon dispersion is derived from the Heisenberg model, with exchange interaction values taken in the so-called strong coupling limit ($t \ll U$). This approximation appears valid for the La_2CuO_4 compound, although the U/t ratio in this material is approximately equal to 8 (Table I), which already doesn't strictly correspond to the $U/t \gg 1$ regime. The local Coulomb interaction in SrCuO_2 is even smaller ($U/t = 6.25$), making the strong coupling approximation for exchange interactions definitively invalid in this case.

Authors: We thank the referee for this comment. Our reply has several layers. First, as also clearly mentioned by referee 2, SrCuO_2 is a Mott insulator, in other words an insulator driven by strong local interactions (see e.g., Fig. R1 for ARPES evidence). As discussed above, a rule of thumb is that Mott physics emerges when $U > 8t$. Yet, in the literature values of $U/t < 8$ have been reported even for La_2CuO_4 . Mott physics is therefore not strictly bound to the condition $U > 8t$.

It is already established that cuprate compounds without apical oxygen have a lower Coulomb interaction U [see e.g. Nat. Phys. **6**, 574 (2010) & Sci. Rep. **6**, 33397 (2016)]. We and others therefore find that U/t is smaller in SrCuO_2 compared to, for example, La_2CuO_4 . We agree with the referee that at sufficiently low U/t , the Heisenberg projection of the Hubbard model (strong-coupling approximation) is no longer valid. In fact, the model is self-predicting its own validity. We indeed find that SrCuO_2 is right at the border, where applicability breaks down. This was and is one of our conclusions. In the revised manuscript, we emphasize this fact more clearly.

Firstly, in a strongly correlated metal regime, the exchange interactions have a much more complex form (see, e.g., [PRB 94, 115117 (2016); PRL 121, 037204 (2018); PRB 105, 155151 (2022)]). Secondly, the perturbation expansion in terms of t/U does not apply here, so one would likely need to account for rather long-range and multi-spin exchange interactions (see, e.g., discussions in [RMP 95, 035004 (2023)]). In this regard, I am wondering if the discrepancy between the fit and the actual data simply arises from an inappropriately considered Heisenberg model, unrelated to quantum fluctuations.

Authors: We would like to point out that ARPES measurements on the same series of samples show that the spectral function of undoped SrCuO₂ [PRL 109, 267001 (2012) & PRB 92, 035149 (2015)] is essentially the same as the other undoped parent cuprates, such as La₂CuO₄ [PRL 95, 227002 (2005)] or Ca₂CuO₄Cl₂ [PRL 93, 267002 (2004)] — see Fig. R1. It is our opinion then that the experimental results justify that SrCuO₂ is still in the Mott insulator rather than the correlated metal regime. As such, the Hubbard model and the perturbation in terms of t/U should still be a valid starting point to describe the physics of these systems. In this context, we evaluate the strength of quantum fluctuations and compare across different cuprate Mott insulators. We stress that our conclusions are not extended beyond the regime of Mott insulators.

We thank the referee for bringing up this important point. In the revised manuscript (page 2, last paragraph of the introduction), we have now underlined these experimental facts evidencing the insulator nature of undoped SrCuO₂.

Fig. R1. Comparison of ARPES results on Sr_{0.99}La_{0.01}CuO₂ (left), La₂CuO₄ (middle), and Ca₂CuO₄Cl₂ (right). The comparable spectra functions demonstrate the characteristics of an insulating state due to local interactions [PRB 92, 035149 (2015), PRL 95, 227002 (2005), and PRL 93, 267002 (2004)].

In my opinion, a clear demonstration that the observed effect is indeed due to quantum fluctuations would be the fact that the magnon dispersion is not reproduced by solving the classical Heisenberg model but is instead reproduced by solving the corresponding quantum Heisenberg model. Alternatively, the authors could obtain magnon dispersions by directly solving the Hubbard model, which, for a single-band case,

can be done using diagrammatic extensions of the dynamical mean-field theory [RMP 90, 025003 (2018)]. The spectral function for magnetic fluctuations can be obtained by calculating the spin susceptibility (see, e.g., [PRL 107, 137007 (2011); EPL 122, 57001 (2018); npj Quant. Mater. 3, 54 (2018); Commun. Phys. 2, 163 (2019)]). The results can then be compared to the magnon dispersion deduced from the Heisenberg model with proper exchange interactions.

Authors: We agree that comparison between the spin fluctuation spectra obtained by directly solving the Hubbard and the Heisenberg models would provide more insights into the effect of quantum fluctuations on the spin susceptibilities. However, this theoretical approach is beyond the scope of current work. The key result of our work is that we provide an effective approach to experimentally characterize the strength of correlation and quantum fluctuations, and by doing so identify a cuprate compound with correlations at the border of the Mott insulating regime. We thank the referee for this insightful comment. In the revised manuscript, we have cited these references and pointed out this potential direction for future studies.

Small points:

It would be helpful if the authors mention the energies or colors corresponding to the four components when discussing the low-energy part of the RIXS spectra (first paragraph on the right side of page 2). Otherwise, it is difficult to identify the bi-magnon peak in Fig. 1.

Authors: We thank the referee for the suggestion. We have implemented this in the revised manuscript. In addition, we now indicate each fitting component in the figure legend directly.

The estimated value of the next-nearest-neighbor hopping $|t'/t|=0.4$ is rather large compared to what is usually used to describe cuprates ($|t'/t|_{\text{max}} = 0.3$). It would be helpful if the authors could compare the obtained model parameters to those obtained in density functional theory (DFT) calculations.

Authors: Parametrizing DFT band structure using a two-band tight binding model, $|t'/t| = 0.41$ for Hg1201 and $|t'/t| = 0.35$ for La_2CuO_4 are found in PRL **105**, 057003 (2010). We have added this information (including relevant citation) to the revised manuscript.

In the Heisenberg model, the authors consider only particular types of exchange interactions, neglecting, e.g., the three-spin interaction, which should be non-zero in the case of an incommensurate spin order. Is there a specific reason for this, and can this interaction be added to the model?"

Authors: For the effective spin Hamiltonian we used, there are no three spin contributions [PRB **85**, 100508 (2012)]. This is because in the case of three-site loops, for each hopping process going around the loop, there will be another going in the other direction which will give the same contribution but with an opposite sign of fermionic origin. Therefore, the net contribution of the three-spin interaction is always zero.

Reviewer #2 (Remarks to the Author):

In their manuscript, Wang et al present a RIXS investigation of the infinite layer SrCuO₂ (SCO), a Mott insulator distinguished in the cuprate realm by the absence of apical oxygen. The authors specifically focus on magnetic excitations, measuring them at different crystal orientations (H0, HH, and azimuthal). It is observed that the measured energy dispersion in q of the magnetic excitations cannot be fitted using the Hubbard model. This model, which effectively sets potential and kinetic energy scales, aligns well with La₂CuO₄, a cuprate parent compound characterized by the presence of apical oxygen. According to the authors, this discrepancy can be reconciled by considering a momentum-dependent normalization factor Z , intricately linked to quantum fluctuations. The conclusions drawn from the fit parameters, notably the substantial value of the exchange coupling, lead the authors to assert that in this system, the antiferromagnetic ordered Néel state is in close proximity to breakdown conditions. This implies the potential stabilization of novel magnetic ground states under extreme conditions, a hypothesis explored further through Monte Carlo simulations.

The significance of the study is underscored by the intense investigation of infinite-layer cuprates in recent times, revealing unexpected phenomena that could contribute to unraveling the physics behind HTS cuprates.

The dataset underpinning the paper is commendable in terms of quality, and the authors extraction of the magnon dispersion is rather convincing.

Authors: We thank the referee for the precise summary of our manuscript and the positive comments that recognize the significance and quality of our study. Below, we respond to the referee's comments and suggestions point-by-point.

The different scenarios occurring for SrCuO₂ and La₂CuO₄ are evident. On the contrary, the connection between the unconventional magnon dispersion of SCO and the presence quantum fluctuations looks weaker. The argument relies exclusively on the chosen fit, wherein the relationship between "measured" and "expected" magnon energy is mediated by a quantum-fluctuation-driven normalization factor. While there is a temptation to draw the conclusion that quantum fluctuations are steering the unconventional behavior, the current evidence suggests more of a suggestive hint than an established fact.

Author: We would like to stress that in our modeling, the magnon quantum renormalization factor is not a fitting parameter that simply "reconciles" the mismatch between the "measured" and "expected" magnon energy. It is calculated (self-consistently) by considering the magnon-magnon interactions as perturbations in the spin-wave theory, given the parameters of the Hubbard model. In the Hubbard model, stronger magnon quantum renormalization is a natural outcome of the decrease in U/t as revealed in PRB **79**, 235130 (2009) & PRB **85**, 100508 (2012), consistent with our calculations shown in Fig. 4.

As mentioned in our reply to referee 1, it has been shown that cuprates without apical oxygen have a weaker correlation strength [Nat. Phys. **6**, 574 (2010) & Sci. Rep. **6**, 33397 (2016)]. Our conclusion that SrCuO₂ has a smaller U/t compared to La₂CuO₄ is consistent with these results.

Therefore, our conclusion on the quantum renormalization effect is not only justified by the successful description of the magnon dispersions of La_2CuO_4 and SrCuO_2 , but also the ability to determine electronic parameters that are self-consistent and accordant with established results.

To substantiate the assertion of the presence of quantum fluctuations in the system, a more thorough comparison with previously collected similar data and an increased dataset are necessary. The manuscript conveys a strong statement on this matter, commencing from the title, and a more robust foundation is needed.

A more detailed comparison with previously measured existing data and a more complete dataset would be required to claim the presence of quantum fluctuations in the system, a rather strong statement appearing in the manuscript starting from the title.

Going into specifics, the observed behavior presented by the authors in Fig. 2 bears a striking resemblance to that previously documented in CCO in Ref. 13. There, the magnon dispersion could not be fitted using the nearest-neighbour Heisenberg model, differently than LCO. However, the fit is rather accurate using a one-band Hubbard model or, even better, the phenomenological linear spin-wave Heisenberg model with four nearest-neighbour coupling parameters. Notably, already with a one-band Hubbard model, the discrepancy between LCO and CCO can be reconciled by using a q -independent normalization factor Z and a ring exchange value in CCO, about 5 times larger than in LCO. This prompts a series of questions: What distinguishes the fit in Ref. 13 from the fit employed by the authors of the present manuscript? How do the authors motivate their choice with respect to what has been done in prior RIXS experiments? Is there a compelling physical rationale justifying their selected approach as the most appropriate?

Authors: We thank the referee for the suggestion to compare our fitting model and the ones used in Ref. 13 (Ref. 17 after revision) on CaCuO_2 . The magnon dispersions are indeed very similar. The one-band Hubbard model used in Ref. 13 considers a constant renormalization factor Z_c taken in the large U/t limit. As shown in Fig. 3d of Ref. 13, a fit using this model (dashed line) deviates significantly from the experimental dispersion. More importantly, as revealed by our comparative results on LCO and SCO, the assumption of constant Z_c is no longer valid for the latter. This leads to a systematic underestimation of U as mentioned by the authors of Ref. 13. To substantiate our conclusions, we analyzed data on SCO and CCO using both models (Figs. R2 and R3, new Supplementary Figs. 2 and 3). A consistent quantum renormalization effect is found to describe both data — as shown in revised Fig. 4. Although the phenomenological Heisenberg model with four effective exchange coupling parameters also describes the magnon dispersion satisfactorily, it does not provide any insight on the correlation strength which is a key information of our work.

Fig. R2 (New Suppl. Fig. 2). Magnon dispersions in SrCuO_2 and CaCuO_2 . (a-c) display the magnon dispersions on SCO (blue open circles) and CCO (yellow open squares) and the respective fits (solid curves) using the Hubbard model as described in the manuscript. The momentum trajectories are shown in (d) and (e) for CCO and SCO, respectively. Data on CCO are adapted from Nat. Phys. **13**, 1201 (2017).

Fig. R3 (New Suppl. Fig. 3). Comparison of fits to magnon dispersion in SCO using t - U Hubbard model with a constant Z_c (blue dashed) and the t - t' - t'' - U model with a momentum-dependent Z_c evaluated from the magnon-magnon interaction (blue solid).

On the experimental front, additional data may be indispensable to adequately substantiate the claim of quantum fluctuations in the system. For instance, if these fluctuations bear a quantum nature, shouldn't we observe a temperature dependent trend? A robust examination of the temperature evolution of the measured magnon dispersion would serve as a stringent test. One would reasonably expect that, with increasing temperature, the "measured" and the "bare" magnon energy should converge, signifying the waning influence of quantum fluctuation contributions as the temperature is raised.

Authors: The referee is right that the temperature evolution of a quantum effect may reflect its nature. However, the energy scale of the magnon renormalization is of the order of $0.1J$. Theoretical studies on spin-wave thermodynamics have shown that for 2D square-lattice Heisenberg antiferromagnets, the quantum renormalization has little temperature dependence for $T < JS(S+1)$, which corresponds to 1000 K for the high- T_c cuprates with $J \sim 1500$ K [PRL **61**, 617 (1988) & Kaganov *et al.*, Spin Waves and Magnetic Excitations (1988)]. Indeed, neutron scattering measurements on La_2CuO_4 [PRL **86**, 5377 (2001)] and CFTD [PRL **87**, 037202 (2001)] observed marginal differences between the spin-wave dispersions below $\sim 0.2 T/J$ (see Fig. R4). Another recent neutron scattering and quantum Monte Carlo study on CFTD [J. Phys.: Condens. Matter **32**, 374007 (2020)] shows that the magnon zone boundary anomaly survives up to $\sim 0.5 T/J$, which for high- T_c cuprates corresponds to $T > 700$ K. Therefore, it would be difficult to conclude the quantum nature of the magnon renormalization by observing thermal smearing with currently available experimental conditions for RIXS (≤ 300 K).

Fig. R4. Temperature evolution of spin-wave quantum renormalization factor measured on 2D square-lattice antiferromagnet cuprate CFTD [PRL **87**, 037202 (2001)]. Little temperature effect is observed below $0.2 T/J$. For SrCuO_2 , $0.2J \sim 300$ K.

In general, the manuscript would benefit from an improved discussion about the physics entailing these quantum fluctuations. Examining, from instance, their expected evolution with temperature and doping, and exploring their possible connection to a quantum critical point, would significantly enrich the depth and breadth of this manuscript.

Authors: In spin-wave theory, quantum fluctuations describe the occupations of the bosonic modes as perturbations to the Néel state. The ground state can thus be viewed as a Néel state with finite boson density. The corresponding elementary excitation should therefore be considered as a magnon renormalized by its higher-order expansions, i.e., the magnon-magnon interactions. Throughout the manuscript, we now emphasize that quantum fluctuations stem from magnon-magnon interactions, and added in the introduction part the above explanations on the physics entailing the quantum fluctuations. The possibility to reach a strain or pressure induced quantum critical point and its implication is now highlighted in the discussion. The manuscript furthermore concludes by mentioning how doping studies would be of great interest.

Beside this main concern, I list in the following other minor issues I see:

1) Upon evaluating the pole positions of magnon energy in Figure 2c derived from the fit shown in Fig. 1, the dispersion in SrCuO₂ seems notably flat at q values above 0.15 rlu along the (h, h) direction. This region constitutes the primary point of qualitative disagreement with the Hubbard fit employing a constant Z (dashed line in Fig. 3l), predicting an upward curvature in that q range. However, this plateau is not evident in the bimagnon dispersion within the same q range, despite the latter mimicking the magnon dispersion at other q values, as explicitly noted by the authors. In that q range, the bimagnon dispersion displays an upward curvature, consistent with the behavior observed in the magnon data presented in the map of Fig. 2c. It is noteworthy that the Hubbard fit with higher-order terms, utilized for assessing quantum fluctuations, exhibits a mild, albeit opposite behavior in that q range, manifesting a softening of magnonic energy versus q . A more detailed explanation from the authors elucidating the reasons behind these incongruences is needed.

Authors: We appreciate the referee's careful examination of the data. Compared to the single-magnon, the bi-magnon intensity is much weaker resulting in a relatively larger uncertainty in the fitted pole energies — as shown in Fig. 2. Within such an error, it is difficult to conclude whether the bi-magnon dispersion near the (h, h) zone boundary has a plateau reminiscent of the single magnon. Nevertheless, while the Hubbard model with constant Z_c predicts an upward single-magnon dispersion peaking at $(0.25, 0.25)$, our data on the bi-magnon dispersion do not suggest a maximum of the pole energy at $(0.25, 0.25)$. Note that the nature of bi-magnons in cuprates measured with RIXS is still controversial [RPB 97, 155144 (2018), PRB 85, 214527 (2012)]. We therefore did not place too much weight on the details of the bi-magnon excitation and leave it to future studies. To remove ambiguities, in the revised manuscript, we now make it clear that the similarity between dispersions of the bi-magnon and single-magnon is evident only when “away from the Brillouin zone centre”.

2) The magnon dispersion is measured at a constant 2θ , i.e., at variable L . Do the authors expect the L dependence being non-influent at all, Considering the 2D nature of the CuO_2 planes housing magnetic excitations and the infinite layer structure of SCO? A discussion on this point would be useful.

Authors: The referee is right that the 2D nature of the system ensures that the L dependence of magnetic excitations is negligible. A recent RIXS study on the isostructural compound CaCuO_2 shows that the magnon dispersions measured at a fixed 2θ and a fixed L indeed have negligible difference [Fig. 2e of PRX **12**, 021041 (2022)]. We have now added discussion on this point in the method section and thank the referee for the suggestion.

3) Discussing the $(h,0)$ and (h,h) directions in terms of antinodal and nodal directions is always a bit misleading in this context, as these terms evoke aspects of the material electronic structure and the shape of the superconducting gap.

Authors: The referee makes a good point. We have now removed the labeling in terms of antinodal and nodal directions.

4) Figure 1c and 1d suggest that the second-order polynomial approximation used in the fit to eliminate the background, primarily originating from the electron-hole continuum, is somewhat abrupt. It appears nearly energy-independent, differing from zero both at zero energy loss and in the antistokes region. This could potentially impact the determination of the pole of the bimagnon excitations, whose height is approximately as that the background, or at least significantly increasing its uncertainty. A detailed explanation or refinement of this aspect by the authors is encouraged.

Authors: After scrutinizing the data fitting, we found that the deviation between fits and the data near zero energy loss is primarily because the energy width of elastic peak is slightly larger than the instrumental resolution due to the existence of unresolved phonon modes. Therefore, we released the constraint on the elastic peak width and refined the fitting of all raw RIXS spectra. As shown in Fig. R5 and new Fig. 1, this improves the fitting quality near the elastic line. A new Supplementary Fig. 1 is added to summarize the RIXS spectra fitting. We have redone all subsequent analysis and updated Figures 1-4 to ensure consistency.

As the referee expected, the refined fitting leads to a reduced error on the bi-magnon pole energy, while the change on the dispersion is marginal. The extracted modeling parameters of the single magnon are essentially unaffected, except that the fitted U value with a constant Z_c shown in Fig. 3 changed from 2.18 eV to 2.15 eV. We thank the referee for the comment that helps improve the overall quality of the data fitting.

Fig. R5. Comparison of RIXS spectra fitting before (a,b) and after (c,d) revision. In the fitting before (a,b), the energy width of elastic peaks (grey shaded area) was constrained by the instrumental resolution, which is now released in the refined fitting (c,d). The extra fitting parameter improves the fit around the elastic line while having a very marginal influence on the magnon excitations.

5) In Figure 2a and 2b, there is a conspicuous suppression of spectral weight at (0.5,0). Given this observation, it is perplexing why the error bars determined for the pole positions of magnons and bimagnons appear almost unaffected by such a signal drop.

Authors: The error bars represent fitting uncertainty of the RIXS spectra. They are not directly determined by the RIXS intensity but rather the identification of each peak's profile. For example, at low Q , the RIXS intensity is large but the magnon excitations are closer to the elastic line, such that the fitting has a larger uncertainty. In our analysis, the magnon and bi-magnon components are determined by globally fitting the spectra obtained with π and σ incident light polarisations. As shown in Fig. R6(a,c), the single-magnon peak remains well-defined near the (0.5, 0) zone boundary with π polarisation, despite its weakened intensity. Meanwhile, the suppression of the single-magnon peak with σ polarisation facilitates the identification of the bi-magnon peak — see Fig. R6(b,d). As such, the uncertainties in determining the pole positions are not strongly affected.

Fig. R6. Fittings of RIXS spectra obtained with π (a,c) and σ (b,d) incident light polarisations. (c,d) display the enlarged figure of spectra at (0.463, 0). The solid lines are fits to the data. The grey dashed lines mark the zero energy loss. Blue and purple shaded areas indicate the fitted components of single-magnon for π and σ polarisations, respectively. Orange shaded areas indicate the fitted components of the bi-magnon.

6) In the methods, for clarity the instrumental energy resolution should be specified in terms of FWHM.

Authors: The instrumental energy resolution is now given in terms of FWHM in the revised manuscript. We also corrected a typo on the previously given energy resolution (HWHM = 59 meV instead of 68 meV). This typo only appeared on the main text without affecting any data analysis.

7) An even minor color coding issue is noted in Figures 3 and 4. The lines, data points, and dashed circles referring to LCO/STO and LCO/LSAO are respectively represented in green and pink in both figures. Conversely, for SCO, they are blue in Figure 3 and orange in Figure 4. To maintain consistency, it is suggested that one color be chosen for both figures in this instance as well.

Authors: We thank the referee for this suggestion. We have now changed the marker color for SCO in Fig. 4 to keep consistency throughout the manuscript.

List of additional changes:

All new text in the revised manuscript and Supplemental Information is indicated with blue colour. Besides, we have made the following changes.

1. We have analyzed RIXS data on CaCuO_2 in new Supplementary Fig. 2. The results of analysis are included in Figure 4 and Table 1. Corresponding discussions are added to Page 4, right column.
2. A new Supplementary Fig. 3 comparing the fits using different Hubbard models is added to the Supplemental Information.
3. All RIXS raw spectra fitting are refined as suggested by the referee. A new Supplementary Fig. 1 is added to summarize the RIXS spectra fitting. We have redone all subsequent analysis leading to updated Figures 1-4. No previous conclusions are affected due to this fitting quality improvement.
4. In Page 2, line 156, we include a possible scenario that may explain the fourth feature of the observed dd excitations and cite the corresponding reference.
5. In Figures 2 and throughout the manuscript, the labellings in terms of “antinodal” and “nodal” are removed.
6. Two coauthors Y. Chan and L. Martinelli are included due to their contributions to the additional data analysis.

REVIEWERS' COMMENTS

Reviewer #1 (Remarks to the Author):

I would like to express my gratitude to the authors for carefully considering my comments and improving the manuscript accordingly. I find the authors' responses to my questions convincing. Therefore, I recommend publishing the revised version of the manuscript in Nature Communications.

Reviewer #2 (Remarks to the Author):

The authors have implemented significant revisions to the manuscript, taking careful consideration of the feedback provided in both my previous report and that of the other referee. Consequently, the paper has achieved greater clarity.

Both the title and the presentation of the analysis now exhibit a closer alignment with the data.

The comparison with CCO, which shares strong similarities with SCO, strengthens the authors' findings; I suggest emphasizing this aspect from the outset.

Regarding the temperature dependence of the magnon dispersion, I appreciate the authors' argument, although it's worth noting that there are RIXS facilities (such as ID32 at ESRF) capable of reaching temperatures up to 600 K. Despite this, I still believe that conducting a RIXS test, even at a temperature of 300 K, and demonstrating even a slight change in the magnon dispersion in the expected direction, would have provided a highly compelling benchmark to incontrovertibly validate the authors' interpretation. It is regrettable that this aspect will not be included in the current manuscript. Nonetheless, I recommend that the authors incorporate the argument presented in their response letter regarding the limitations in probing the quantum nature of magnon renormalization by observing changes within a temperature range of a few hundred kelvin.

Apart from these points, I have no further comments and confidently recommend the paper for acceptance in Nature Communications.

RESPONSE TO REVIEWER COMMENTS

Reviewer #1 (Remarks to the Author):

I would like to express my gratitude to the authors for carefully considering my comments and improving the manuscript accordingly. I find the authors' responses to my questions convincing. Therefore, I recommend publishing the revised version of the manuscript in Nature Communications.

Authors: We thank the reviewer for recommending the publication of our manuscript in Nature Communications.

Reviewer #2 (Remarks to the Author):

The authors have implemented significant revisions to the manuscript, taking careful consideration of the feedback provided in both my previous report and that of the other referee. Consequently, the paper has achieved greater clarity. Both the title and the presentation of the analysis now exhibit a closer alignment with the data.

Authors: We thank the reviewer for reviewing our paper again and the positive comments.

The comparison with CCO, which shares strong similarities with SCO, strengthens the authors' findings; I suggest emphasizing this aspect from the outset.

Authors: We have included additional text in the last paragraph of the introduction to emphasize this aspect.

Regarding the temperature dependence of the magnon dispersion, I appreciate the authors' argument, although it's worth noting that there are RIXS facilities (such as ID32 at ESRF) capable of reaching temperatures up to 600 K. Despite this, I still believe that conducting a RIXS test, even at a temperature of 300 K, and demonstrating even a slight change in the magnon dispersion in the expected direction, would have provided a highly compelling benchmark to incontrovertibly validate the authors' interpretation. It is regrettable that this aspect will not be included in the current manuscript. Nonetheless, I recommend that the authors incorporate the argument presented in their response letter regarding the limitations in probing the quantum nature of magnon renormalization by observing changes within a temperature range of a few hundred kelvin.

Authors: We have incorporated discussions on this point at the end of the discussion section.

Apart from these points, I have no further comments and confidently recommend the paper for acceptance in Nature Communications.

Authors: Finally, we wish to thank both reviewers again. Our manuscript is significantly improved after incorporating all the constructive comments and suggestions. We believe that our manuscript is now ready for publication in Nature Communications.